# A Multi-Channel Attention Network for SAR Interferograms Filtering Applied to TomoSAR

Jie Li [1,2,3,*], Zhiyuan Li [1,2,3], Bingchen Zhang [1,2,3] and Yirong Wu [1,3]

1 Aerospace Information Research Institute, Chinese Academy of Sciences, Beijing 100094, China; lizhiyuan19@mails.ucas.ac.cn (Z.L.); zhangbc@aircas.ac.cn (B.Z.); wyr@mail.ie.ac.cn (Y.W.)
2 Key Laboratory of Technology in Geo-Spatial Information Processing and Application System, Chinese Academy of Sciences, Beijing 100190, China
3 School of Electronic, Electrical and Communication Engineering, University of Chinese Academy of Sciences, Beijing 101408, China
* Correspondence: lijie195@mails.ucas.ac.cn

**Abstract:** Tomographic synthetic aperture radar (TomoSAR) is an advanced synthetic aperture radar (SAR) interferometric technique that can retrieve 3-D spatial information. However, the performances of 3-D reconstruction could be degraded due to the noise in interferograms, which makes the filtering crucial before the tomographic reconstruction. As known, filters for single-channel interferograms are common, but those for multi-channel interferograms are still rare. In this paper, we propose a multi-channel attention network to denoise the multi-channel interferograms applied for TomoSAR, which is built on the basis of multi-channel attention blocks. An important feature of the block is the local context mixing before the computation of attention maps across channels, which explores the intra-channel local information and the inter-channel relationship of the multi-channel interferograms. Based on this architecture, the proposed method can effectively filter the noise while preserving the structures in interferograms, thus improving the performance of tomographic reconstruction. The network is trained by simulated data and the promising results of both simulated and real data validate the effectiveness of our proposed method.

**Keywords:** tomographic synthetic aperture radar (TomoSAR); interferogram; multi-channel attention network; convolutional neural network (CNN)

## 1. Introduction

Synthetic aperture radar (SAR) is a remote sensing technology that can deliver SAR data with high spatial resolution. The emergence of modern SAR systems has led to a breakthrough in 3-D surface imaging, particularly. Tomographic synthetic aperture radar (TomoSAR) is widely used for providing 3-D spatial information based on the multi-channel complex interferograms [1,2]. The reflectivity profiles of observed scatterers along the elevation direction can be reconstructed by the multi-channel interferometric synthetic aperture radar techniques.

However, the complex interferograms are corrupted by speckle [3], bringing about the disturbances and distortions that lead to the high number of errors and standard deviations in the interferograms [4,5]. Furthermore, the reflectivity reconstruction by TomoSAR would be degraded, which shows the significant biases and discontinuities in localization [6]. For this reason, the precise estimation of the noisy SAR interferograms becomes a fundamental and crucial step to ensure the correctness of reconstruction in tomographic SAR imaging [7–9].

In recent decades, numerous filtering methods have been proposed for the restoration of interferograms [10]. The multi-look filter is a traditional method that simply performs a moving average to estimate the variation of local pixel pattern [11]. It is still widely used nowadays because of the straightforward operation. The fundamental limitation of

the multi-look filter comes from the loss of spatial resolution and details of the interferograms, due to the same smoothing effect being equally applied to the interferograms of homogeneous and heterogeneous regions. Consequently, classical filters working in both spatial and transform domains have emerged. For example, Lee et al. [12] proposed an adaptive filtering for interferometric SAR denoising in the original spatial domain. The directional window is locally selected among eight edge-aligned windows based on the local gradient of the amplitude images according to the principle of Lee filter, which preserves edge structures well. Moreover, there are also some works [6,13] that estimate the complex interferometric SAR data by Markovian modeling. In addition to studies in the spatial domain, Goldstein and Baran filters [14,15] have been proposed to estimate the dominant component from the local power spectrum of the signal in the frequency domain. Inspired by the success of wavelet-domain methods on natural image restoration, the filter in [16] was proposed to separate the phase information and noise easily in the wavelet domain, which seems to preserve a good spatial resolution and have high computational efficiency [17]. However, these methods have not overcome the neighbor-connection limitation. Recently, the nonlocal filter [18,19] considers the pixels that are far apart, which are selected by exploiting the redundant patterns to combine for the estimation of each given pixel. Moreover, this technique uses a probabilistic criterion based on the complex interferometric SAR data that surround two given patches to select suitable samples. In [20], the phase noise filtering formulations with the norm regularizers are established. In [21], a novel convolutional sparse coding method with the prior knowledge of coherence was proposed. This optimization model is not only capable of reducing noise in regions with continuous phase changes, but also of preserving the phase details prominently.

Recently, deep-learning-based methods, especially deep convolutional neural network (CNN) techniques, have shown their dominant performance in the past few years on different visual-related tasks, including image restoration [22]. Milestone works based on CNN have been introduced in the single-channel interferogram restoration, showing their ability to outperform the conventional algorithms [23–26]. In [27], the residual learning strategy combined with a densely connected feature extractor was adopted to conduct the filtering of interferograms. In [28], Φ-Net was proposed for the joint estimation of interferometric SAR phase and coherence. This network has the capability to perform blind denoising and preserve high-frequency signal components of interferograms with different noise levels. In [29], the network with a multi-objective cost function was proposed, which contained the residual blocks composed of CNN architecture with a skip connection in the output of the last.

TomoSAR exploits multi-channel interferograms to retrieve 3-D spatial information. To denoise the multi-channel interferograms, the concepts of classical filters are migrated to deal with multiple interferograms such as the multi-look filter [30]. A successful trend is the nonlocal framework, which has been utilized to deal with the multi-channel interferograms from tomographic SAR data [7,8,31]. These filters estimate the parameters of interferograms based on the multi-dimensional probability density function (PDF) or covariance matrix, and further improve the performance of tomographic reconstruction.

Clearly, most of the deep-learning-based networks have been widely used in the filtering of the single-channel interferogram, which could effectively mitigate the noise. To exploit the multi-channel information, the attention mechanism has been utilized to attentively select interested intermediate generations that could significantly boost the quality of the final output [32–36]. The attention maps are calculated across feature channels for parallelization and effective representation learning, which have shown state-of-the-art performance on natural language tasks [37], high-level vision problems [38,39], and image restoration tasks [40–42]. The attention block computes the local context via convolution operations and ensures the contextualized global relationships between channels by computing covariance-based attention maps.

In this paper, we propose a novel multi-channel attention network for filtering the multi-channel interferograms applied in TomoSAR. The network is designed on the basis

of multi-channel attention blocks, which consist of pixel-wise aggregation using convolution operations, and cross-covariance computation via channels attention maps. Thus, the proposed network can explore the intra-channel local information and inter-channel relationship to reduce the noise in multi-channel interferograms. Simulated datasets are fed into the network for the training process, and the results of simulated and real SAR images ensure the effectiveness of the proposed method.

## 2. Methodology

### 2.1. Tomographic SAR Interferograms

For a single SAR acquisition, the focused complex-valued measurement $g_m$ of an azimuth-range pixel $(x_0, r_0)$ for the $m$th acquisition at aperture position $b_m$ is the tomographic projection of the reflected signal along the elevation direction in the presence of the noise,

$$g_m = \int_{\Delta s} \gamma(s) exp(-j2\pi\xi_m s)ds + \varepsilon_m \tag{1}$$

with

$$\xi_m = \frac{-2b_m}{\lambda r} \tag{2}$$

where $\gamma(s)$ is the reflectivity function along elevation $s$ with an extent of $\Delta s$. The spatial frequency $\xi_m$ is proportional to the respective aperture position $b_m(m = 1, 2, \ldots M)$. $\lambda$ is the wavelength of radar signals, and $r$ denotes the range between radar and the observed object, respectively. $\varepsilon_m$ stands for the Gaussian white noise.

Based on the preprocessing steps in TomoSAR, the complex interferogram $I$ is defined as the complex conjugate product of the two noisy co-registered single-look complex (SLC) SAR images $(g_{m_1}, g_{m_2})$ with $m_1, m_2 = 1, 2, \ldots M, m_1 \neq m_2$,

$$I = g_{m_1} g_{m_2}^* \tag{3}$$

TomoSAR uses the multi-channel complex interferograms to obtain the reflectivities along elevation direction according to the principle in Figure 1. The elevation information of observed scatterers exists in the interferometric phase and the reflectivity profiles depend on the amplitude of interferograms. However, speckle degrades the quality of interferograms severely, resulting in the noise and distraction of the complex interferograms. This phenomenon definitely leads to the reduction of correlation between the interferometric pair, and large errors in the estimated reflectivity profiles. The correlation quantity is measured as the module of the complex correlation between the two SLCs, which is normally indicated as coherence. Meanwhile, the noise in the multi-channel interferometric phases could degrade the quality of interferograms severely by increasing the standard deviation of the interferometric phases [12,20]. According to the principle of TomoSAR imaging, the standard deviation of interferometric phase noise could definitely lead to a higher standard deviation of tomographic reconstruction.

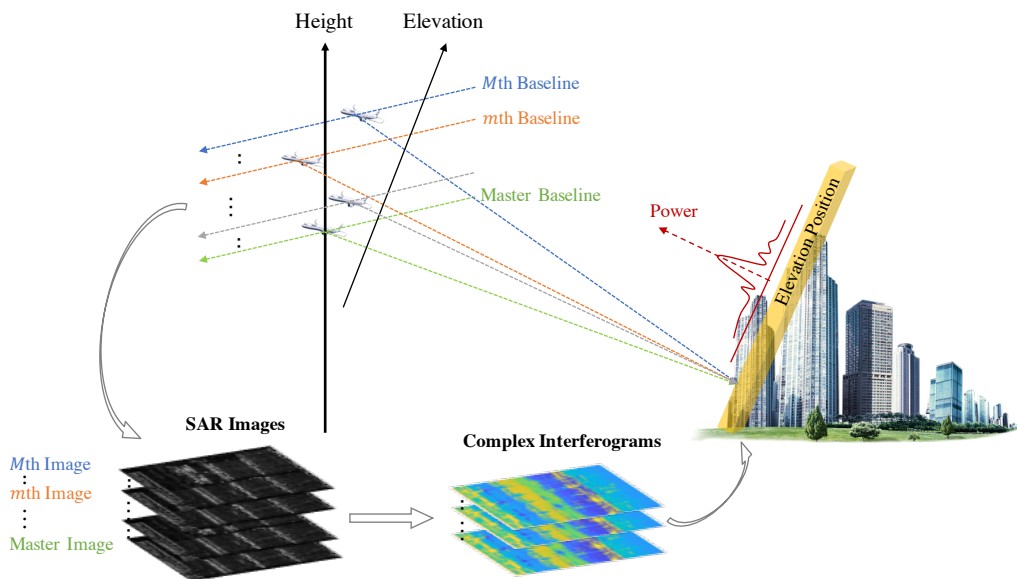

**Figure 1.** The principle of TomoSAR.

## 2.2. *The Proposed Multi-Channel Attention Network*

To conduct the restoration of multi-channel interferograms, we apply the novel multi-channel attention blocks for the cross-channel interferogram guidance task. It can effectively learn the intra-channel local context and inter-channel attention maps to guide the pixel loss for more robust optimization of multi-channel interferograms.

### 2.2.1. Overall Structure of the Proposed Framework

An illustration of the overall network structure is depicted in Figure 2a with reference to [42]. Given the noisy multi-channel interferogram images $\Gamma_{in}$, the proposed network first applies the $3 \times 3$ convolution operations to obtain low-level feature embeddings $\Gamma_{F,0}$. Then, these shallow features pass through a four-level symmetric encoder–decoder and are transformed into deep features. Each level of the encoder–decoder contains multi-channel attention blocks. The encoder hierarchically reduces spatial size by half while expanding channel capacity. The decoder takes low-resolution latent features $\Gamma_{F,l}$ as the input and progressively recovers the high-resolution and clear representations. For feature downsampling and upsampling, we apply pixel-unshuffle and pixel-shuffle operations [43], respectively, and the encoder features are concatenated with the decoder features via skip connections [44] in order to assist the recovery process. All the concatenation operations are followed by a $1 \times 1$ convolution layer to keep the number of channels consistent, except the top one. At Level 1, we let the multi-channel attention block aggregate the low-level image features of the encoder with the high-level features of the decoder, leading to the feature maps with twice the number of channels. It is beneficial in preserving the fine structural details in the restored images by aggregating the low-level image features of the encoder with the high-level features of the decoder. Next, the deep-level features $\Gamma_{F,d}$ are further enriched by the remaining multi-channel attention blocks, and the following convolution is applied to generate residual maps of the features. The number of channels of the output feature map is the same as that of the input layer. Finally, the $3 \times 3$ convolution is applied to transform the obtained feature result to interferogram images. Generally, the top and bottom convolution operations also take into account the different forms of noise in the tomographic SAR interferograms.

To transform features, we apply the feed-forward network with the architecture in Figure 2b. The GELU nonlinearity function and the depth-wise convolution work together to control the information flow through the respective hierarchical levels and allow the feature of each level to focus on the fine details complimentary to the other levels [42].

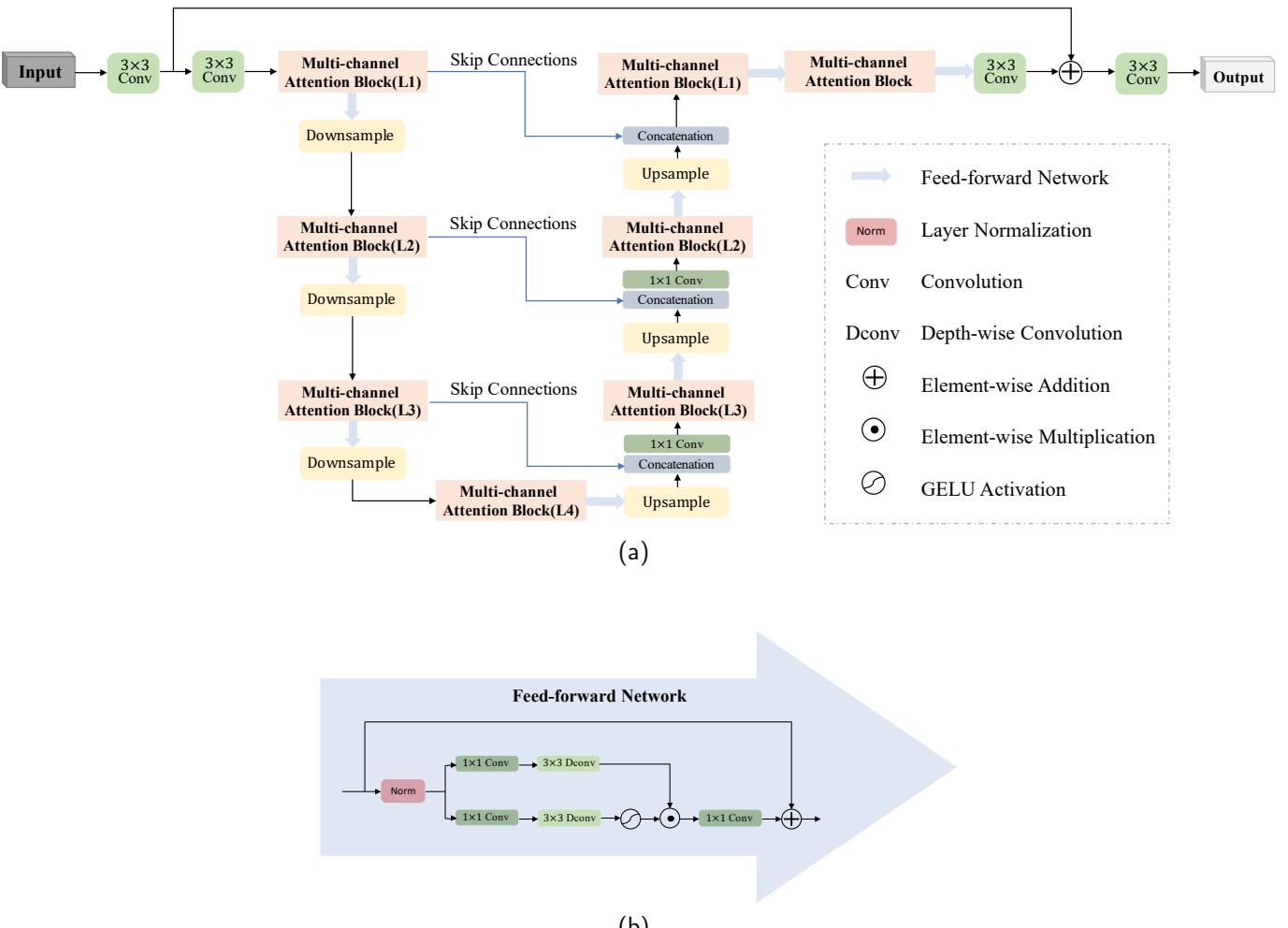

**Figure 2.** (**a**) Overview of the proposed multi-channel attention framework and (**b**) the contained feed-forward network. The proposed network consists of the multi-channel attention block designed for incorporating the feature information of different channels, where L1 in parentheses represents Level 1. The concatenation operation is followed by a $1 \times 1$ convolution to keep the number of channels consistent at all levels, except the top one.

### 2.2.2. Multi-Channel Attention Block

In this section, we describe the core multi-channel attention block of the proposed network, which is shown in Figure 3. One of the main components of utilized blocks is convolution operations to achieve the pixel-wise aggregation, which can help to explore the local context information of interferograms. Another lies in the calculation of channel attention maps to make full use of the cross-covariance information across channels, which can preserve the inter-channel relationship of multi-channel interferograms. Thus, the designed network has the ability to conduct the denoising and suppress the standard deviation of filtered interferograms.

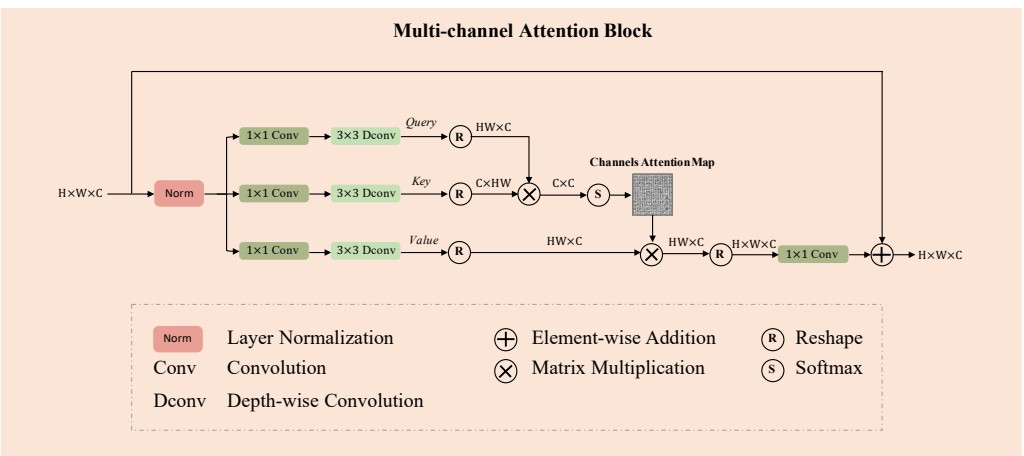

**Figure 3.** The multi-channel attention block in the proposed framework.

With regard to the multi-channel attention block, we design the architecture based on the mechanism of multi-head self-attention [33]. The utilized blocks generate three projections, including query (**Q**), key (**K**), and value (**V**). Each projection consists of the $1 \times 1$ convolution layer and $3 \times 3$ depth-wise convolution layer. The former can aggregate pixel-wise cross-channel information, and the latter can encode channel-wise spatial information. Thus, the obtained projections are enriched with local context, which can be expressed as follows:

- $\Gamma_{F,\mathbf{Q}} = W_d^{\mathbf{Q}} W_p^{\mathbf{Q}} \Gamma_{F,mc}$;
- $\Gamma_{F,\mathbf{K}} = W_d^{\mathbf{K}} W_p^{\mathbf{K}} \Gamma_{F,mc}$;
- $\Gamma_{F,\mathbf{V}} = W_d^{\mathbf{V}} W_p^{\mathbf{V}} \Gamma_{F,mc}$.

where $\Gamma_{F,\mathbf{Q}}$, $\Gamma_{F,\mathbf{K}}$, and $\Gamma_{F,\mathbf{V}}$ represent the query, key, and value projections, respectively. $\Gamma_{F,mc}$ denotes the feature maps after the layer normalization of the multi-channel attention block. $W_d$ denotes the operation of $3 \times 3$ depth-wise convolution in each projection, and $W_p$ denotes the operation of $1 \times 1$ convolution in each projection. Then, we reshape query and key projections so that their product of matrix multiplication can generate the channel attention map with size of $C \times C$, where $C$ is the number of channels of the input in the multi-channel attention block.

The channel attention map is obtained by a channel-wise softmax function, and the intermediate generation of value projection is guided by the channel attention map, to produce the refined result [42]. It has the ability to relate the different channels of a single sequence to calculate the representation of the sequence [32]. The attention map is trained to calculate the cross-covariance information across the channels for effective representation learning, which reflects the influence of each channel on the other channels. The learned channel attention map is utilized to perform channel-wise selection from each intermediate generation by matrix multiplication operation, which explores the information of different channels.

Based on the designed multi-channel attention blocks, the network learns the 2-D local context information, which could be valuable for the structure preservation, while conducting the suppression of noise caused by speckle in interferograms. Meanwhile, the network explores the inter-channel relationship, which makes it possible to reduce the effect of geometrical/temporal decorrelation. Thus, the network could achieve the denoising purpose well and preserve the structure in multi-channel interferograms.

## 3. Experiment

### 3.1. Network Training

We conduct the simulation of multi-channel interferograms for training, based on ground, buildings, roads, and slopes with various shapes and heights. The training dataset

consists of numerous pairs of noisy interferograms and their corresponding reference images. The speckle is introduced into the training dataset [7]. To enhance the robustness of the network, the training dataset is simulated at different levels of SNR, which is regularly distributed between [−10, 20 dB], and the simulated interferograms are conducted with nine channels.

For the optimization of the proposed network parameters, we exploit the $L_2$-norm loss functions. The differences between noisy multi-channel interferograms and their ground truth are considered by minimizing the $L_2$ loss. As the tomographic SAR interferograms are complex, both amplitude and interferometric phase components are fed to the network. The objective of interferograms denoising can be regarded as the constraints of amplitude and interferometric phase information via $L_2$-norm functions.

We carry out the training procedure using Pytorch and the Adam optimizer, and the network is trained for 1000 epochs using the Adam optimizer, with the learning rate set as 0.001. A batch size of 10 is used in all experiments. The proposed model is implemented in the PyTorch package and runs on an NVIDIA 2060Ti GPU with 6 GB RAM. For the dataset, we simulate 29,686 samples to build the training dataset, which are the patches of $128 \times 128$ pixels.

### 3.2. Simulated Data

To evaluate the effectiveness of the proposed network, we choose two simulated scenes to show the performance, which generate different interferometric phase patterns. The simulated target is 80 m in scene.1 and the simulated target is 42 m in scene.2. The two simlated scenes present different fringe structures of the interferometric phases in order to better verify the effectiveness of the proposed method. Both the simulated systems have ten channels in the cross-track direction, leading to the interferograms with nine channels. The effective baseline is uniformly distributed and the overall effective baseline is 2.25 m, and the incident angle is ideally assumed to be 90°, which indicates that it is unnecessary for converting the radar geometry to ground geometry in these simulation experiments. Gaussian white noise is added with SNR = 5 dB. The simulated scenes and their corresponding multi-channel interferograms are shown in Figure 4.

To demonstrate the denoising performance of interferograms, we apply the classical multi-look and nonlocal [8] filters to verify the effectiveness of our proposed multi-channel attention network (MCAN). Meanwhile, we compare our proposed method with GenIn-SAR [45] and Φ-Net [28]. The obtained results are shown in Figures 5 and 6.

Clearly, the unfiltered images are filled with noisy points, resulting in a poor performance in coherence due to the decorrelation effects. The multi-look filter averages $5 \times 5$ pixels in the range and azimuth directions, leading to the suppression of noise, especially in the flat regions of images. The nonlocal filter performs a weighted averaging of similar pixels based on the multi-channel statistical model of interferograms, which has a better performance in denoising than multi-look filter, especially in the fringe structure of the interferometric phases. The statistical characteristics of coherence of the nonlocal filter show an improvement of coherence values, which is essential for the performance of TomoSAR reconstruction. The GenInSAR produces the better results surpassing the nonlocal filter, through learning the data distribution from training datasets, and the Φ-Net designed on the basis of the U-Net architecture exploits the concept of residual learning by mapping the input toward the output, which shows a strong candidate for the generation of high-quality results. Among the deep-learning-based methods, our proposed MCAN can perform better, which utilizes the cross-covariance information across the channels for denoising. The noisy points in interferometric phases are suppressed, and the coherence value is further improved.

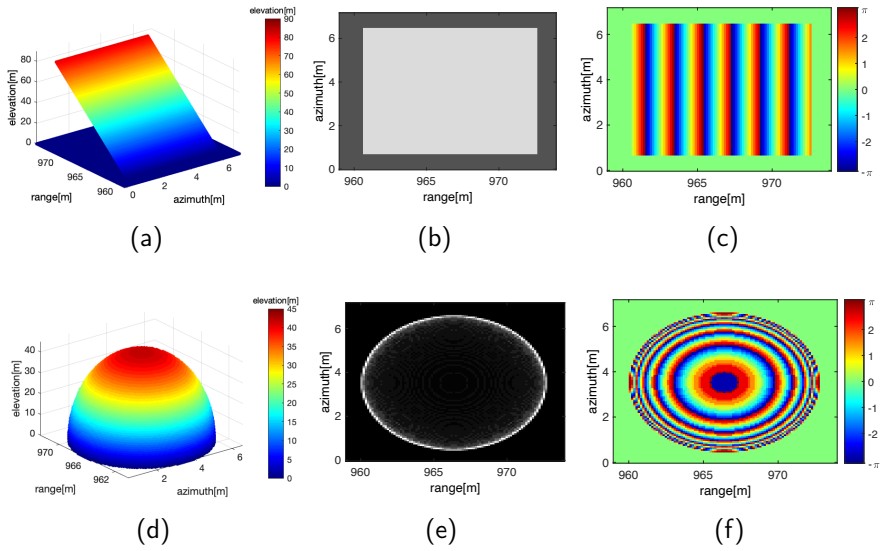

**Figure 4.** The simulated 3-D scenes and their corresponding interferograms. Scene.1: (**a**) 3-D point cloud, (**b**) intensity image of the interferogram, (**c**) interferometric phase. Scene.2: (**d**) 3-D point cloud, (**e**) intensity image of the interferogram, (**f**) interferometric phase.

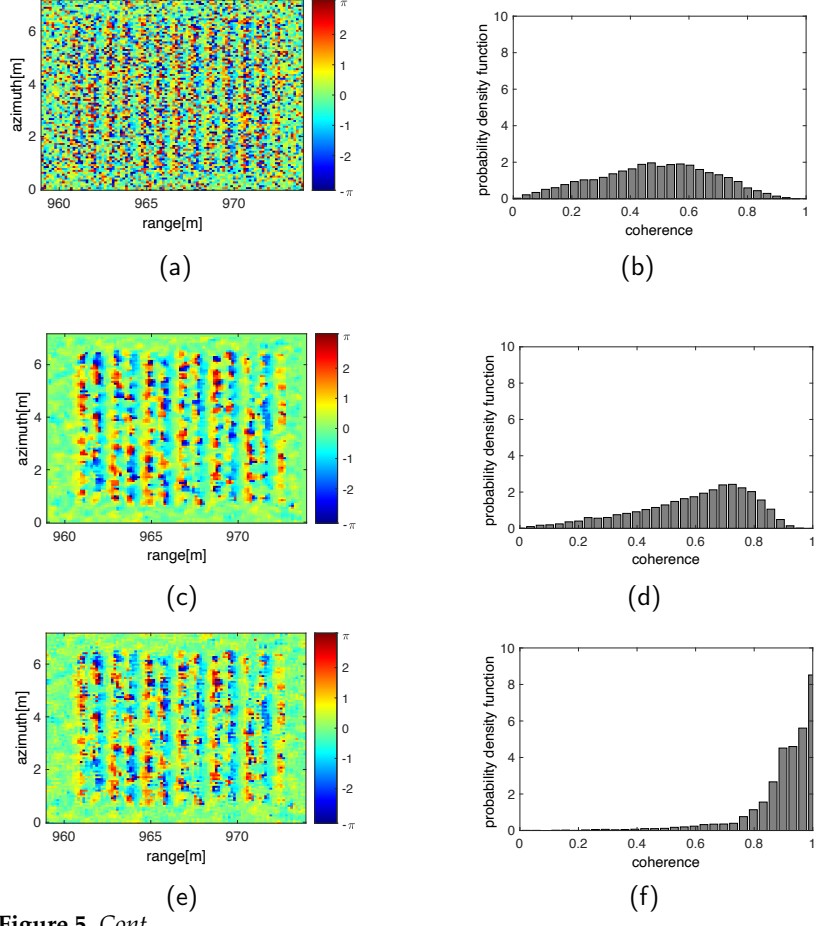

**Figure 5.** *Cont.*

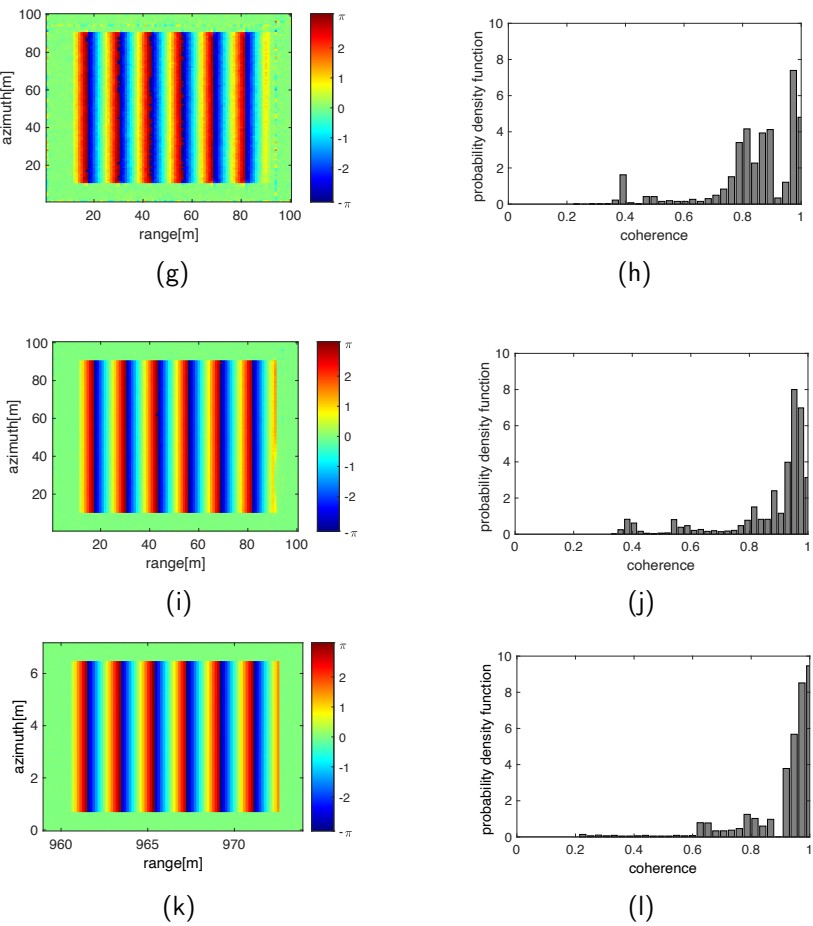

**Figure 5.** Scene.1: The filtered interferometric phases by different methods and their corresponding statistical characteristics of coherence. (**a**,**b**) Unfiltered. (**c**,**d**) Multi-look (5 × 5). (**e**,**f**) Nonlocal. (**g**,**h**) GenInSAR. (**i**,**j**) Φ-Net. (**k**,**l**) Proposed MCAN.

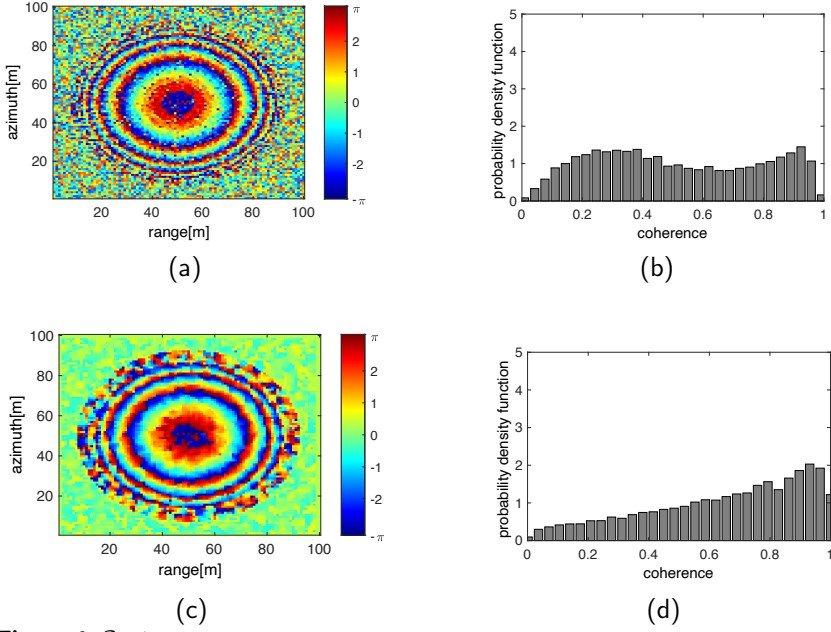

**Figure 6.** *Cont.*

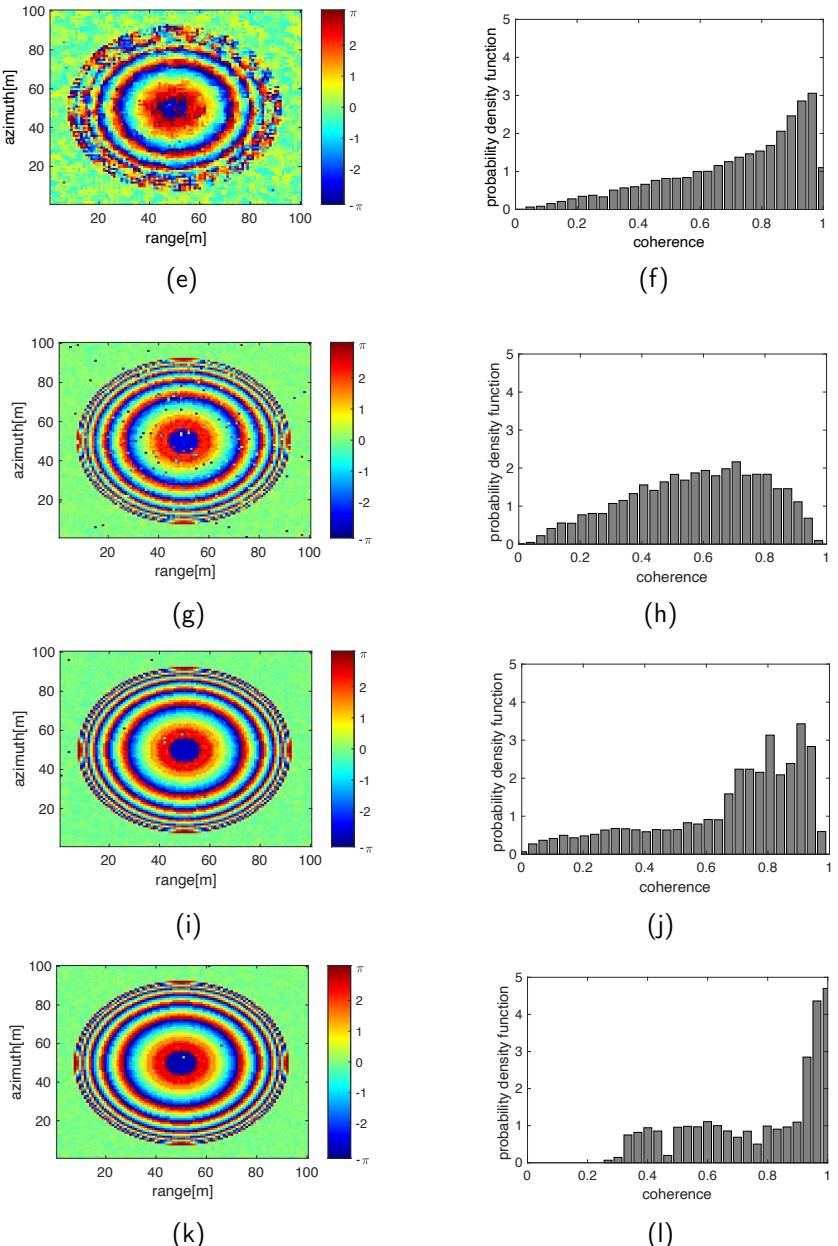

**Figure 6.** Scene.2: The filtered interferometric phases by different methods and their corresponding statistical characteristics of coherence. (**a**,**b**) Unfiltered. (**c**,**d**) Multi-look (5 × 5). (**e**,**f**) Nonlocal. (**g**,**h**) GenInSAR. (**i**,**j**) Φ-Net. (**k**,**l**) Proposed MCAN.

The elevation information is contained in the interferometric phases; thus, the quality of interferometric phases is critical to tomographic reconstruction. Figures 7 and 8 show the statistical answers of different filters on the samples of the two simulated interferometric phases in the middle of the azimuth, where ground truth along the range direction is shown in Figure 9. Clearly, the statistical answer on the unfiltered data presents the large error and standard deviation, which can be reduced by multi-look and nonlocal filters to a certain extent. The multi-look filter suppresses the noise, but keeps the high-standard deviation in the areas with large phase changes, and the proposed method can improve the accuracy and precision of the estimation of interferometric phases while preserving the detailed information of interferograms. From the statistical analysis, the root mean square error (RMSE) for the estimated quantity and the reference one, and the standard deviation (STD) are calculated for the interferometric phase. Focusing on the estimated

interferometric phases, one can note that the best filter is our proposed MCAN with the RMSE metrics of 0.0808 rad and 0.2142 rad in scene.1 and scene.2, respectively. Meanwhile, the corresponding STD metrics are 0.0049 rad and 0.0192 rad. Thus, we find that the estimation of interferometric phases is greatly improved by the proposed method.

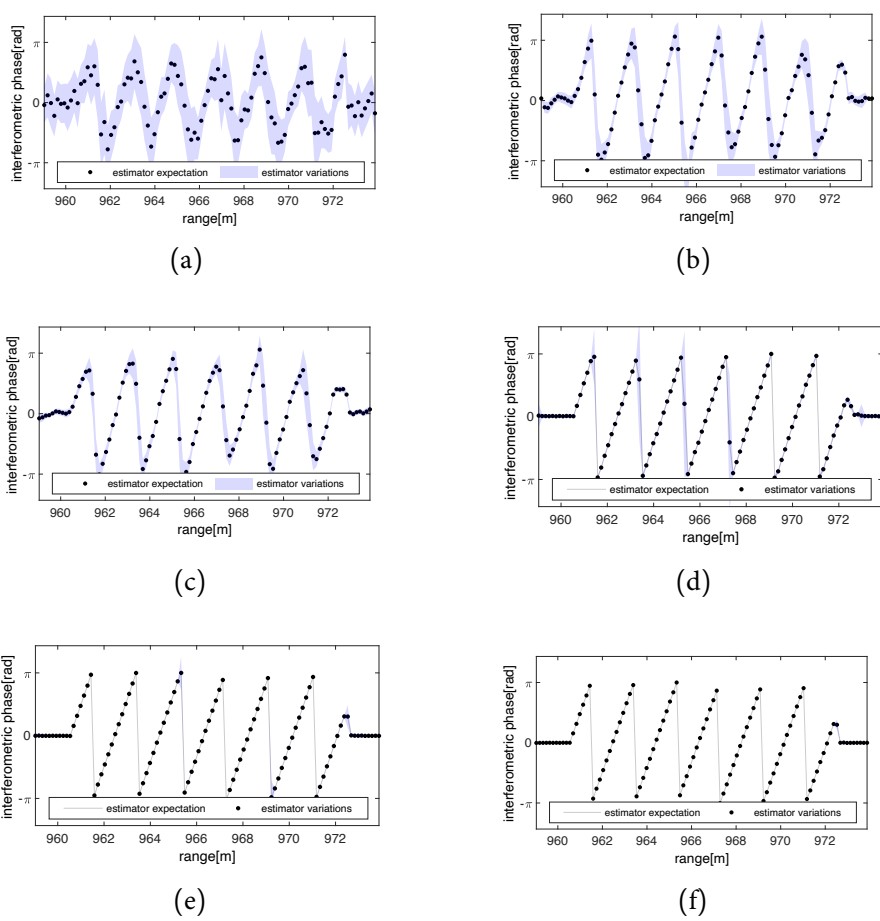

**Figure 7.** Scene.1: The statistical answers for the different filtered interferometric phases. (**a**) Unfiltered. (**b**) Multi-look. (**c**) Nonlocal. (**d**) GenInSAR. (**e**) Φ-Net. (**f**) Proposed MCAN.

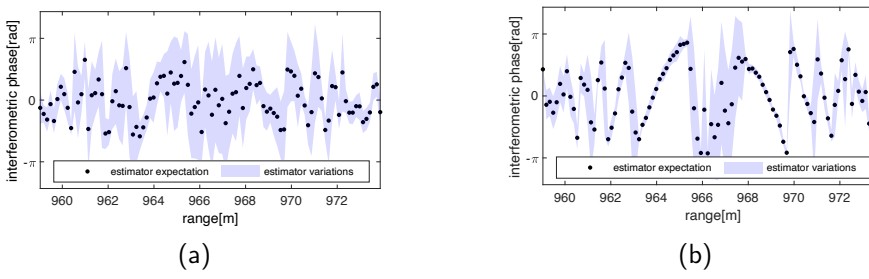

**Figure 8.** *Cont.*

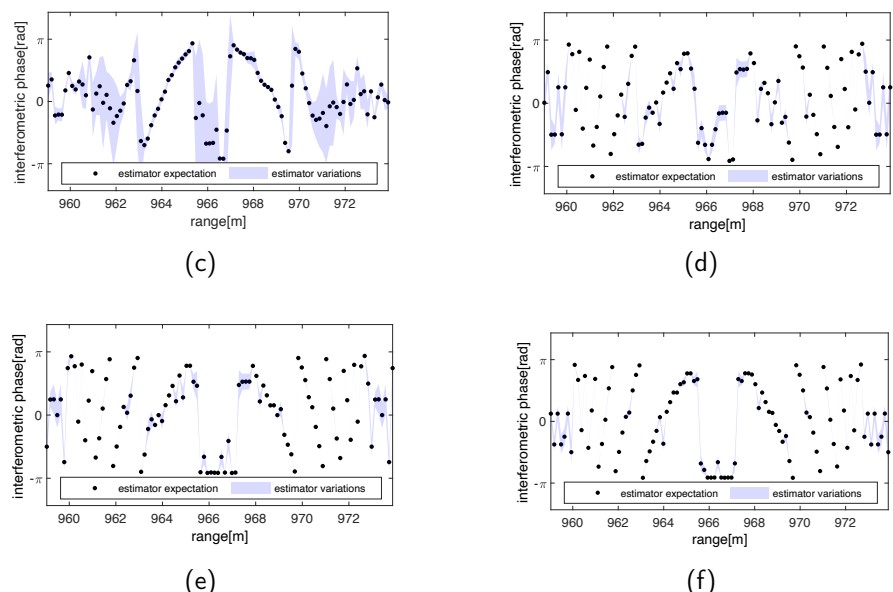

**Figure 8.** Scene.2: The statistical answers for the different filtered interferometric phases. (**a**) Unfiltered. (**b**) Multi-look. (**c**) Nonlocal. (**d**) GenInSAR. (**e**) Φ-Net. (**f**) Proposed MCAN.

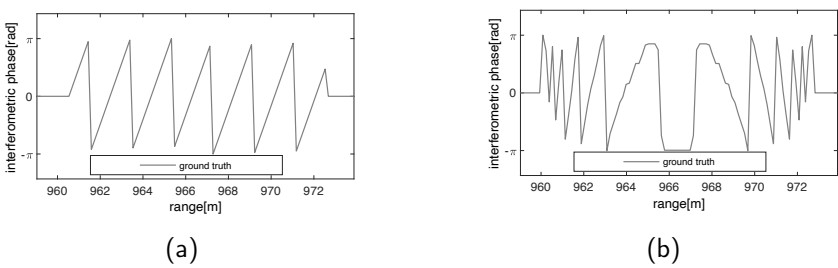

**Figure 9.** The statistical answers for the clean interferometric phases. (**a**) Scene.1. (**b**) Scene.2.

To demonstrate the robustness of noise suppression, we use the simulated data with different levels of SNR in Figures 10 and 11. It is noted that the performance of Φ-Net is relatively good. When SNR = 0 dB, the interferometric phases are severely influenced by noise which indicates the high error and standard deviation. In this case, the Φ-Net conducts the filtering process well and achieves the improved performance with the low standard deviation. Meanwhile, our proposed method has similar filtered results with respect to the error and standard deviation. According to the principle of our proposed method, multi-channel interferograms provide more information. Thus, the filtering performance could perform well by exploiting the cross-channel features, especially in the areas characterized by abrupt phase changes. The performances with SNR = 10 dB are similar to those with SNR = 5 dB in Figures 7 and 8. Generally, the proposed method can achieve the excellent suppression of distraction, leading to the low error and standard deviation.

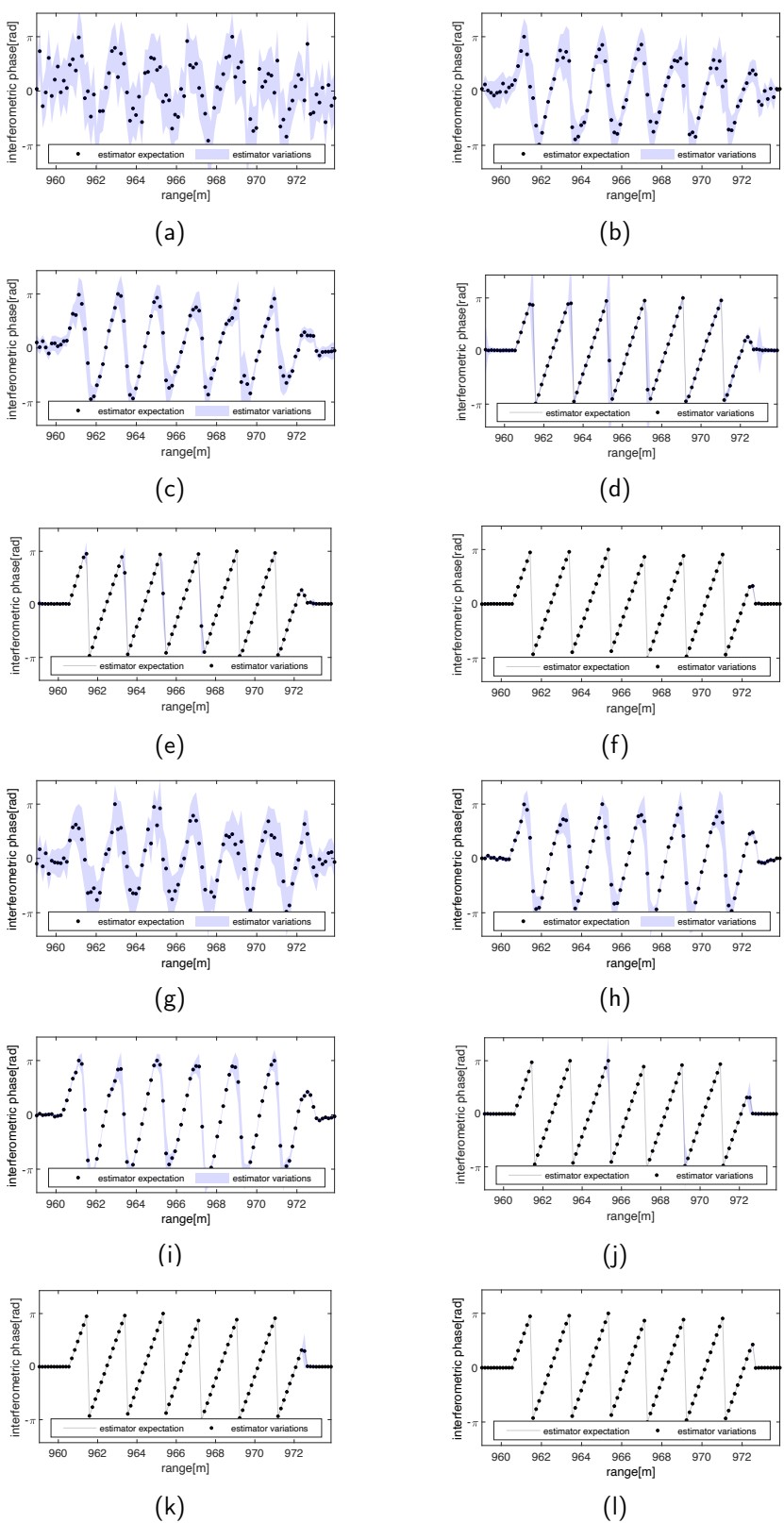

**Figure 10.** Scene.1: The statistical answers for the filtered interferometric phases with different SNRs. (**a**) Unfiltered (SNR = 0 dB). (**b**) Multi-look (SNR = 0 dB). (**c**) Nonlocal (SNR = 0 dB). (**d**) GenInSAR (SNR = 0 dB). (**e**) Φ-Net (SNR = 0 dB). (**f**) Proposed MCAN (SNR = 0 dB). (**g**) Unfiltered (SNR = 10 dB). (**h**) Multi-look (SNR = 10 dB). (**i**) Nonlocal (SNR = 10 dB). (**j**) GenInSAR (SNR = 10 dB). (**k**) Φ-Net (SNR = 10 dB). (**l**) Proposed MCAN (SNR = 10 dB).

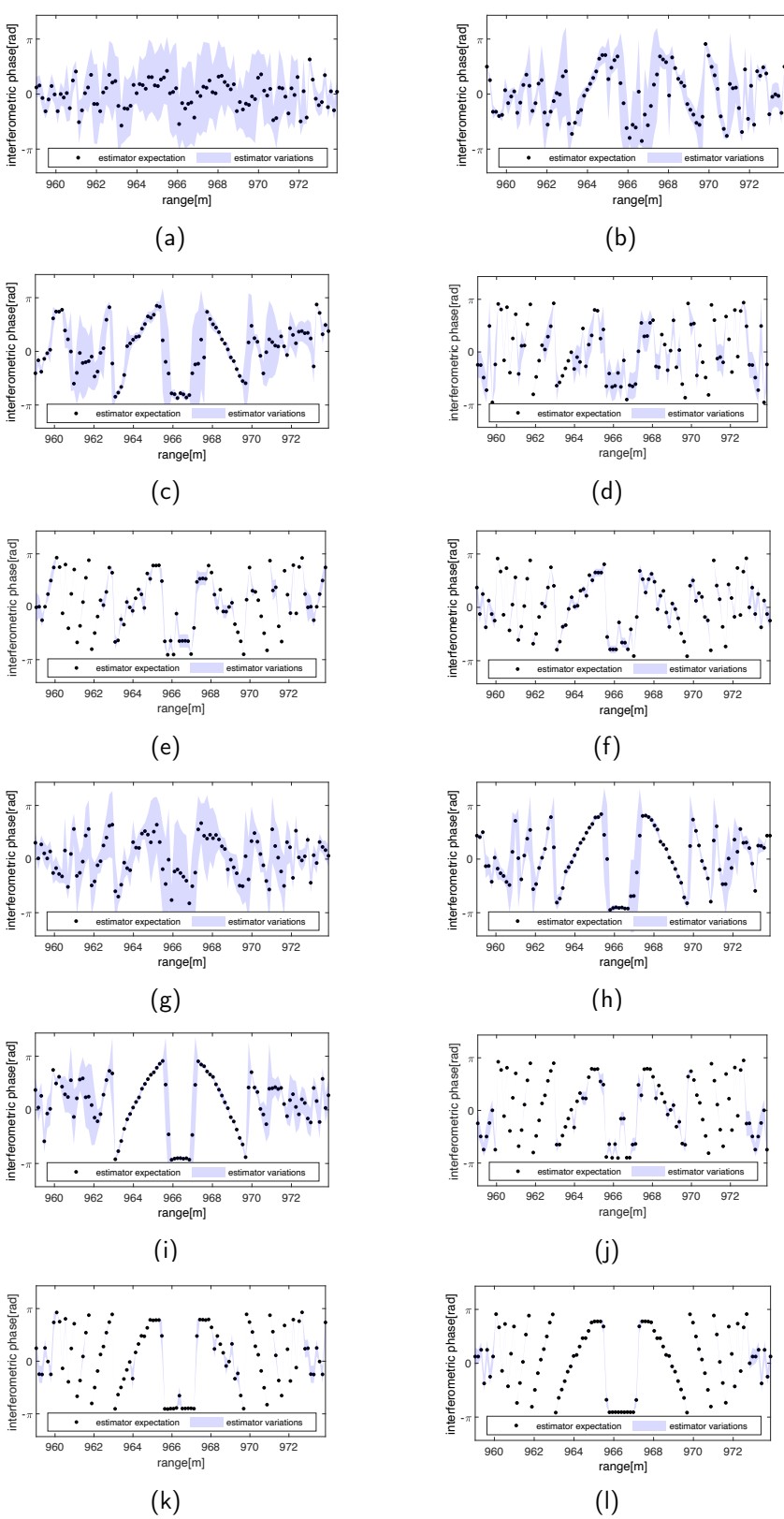

**Figure 11.** Scene.2: The statistical answers for the filtered interferometric phases with different SNRs. (**a**) Unfiltered (SNR = 0 dB). (**b**) Multi-look (SNR = 0 dB). (**c**) Nonlocal (SNR = 0 dB). (**d**) GenInSAR (SNR = 0 dB). (**e**) Φ-Net (SNR = 0 dB). (**f**) Proposed MCAN (SNR = 0 dB). (**g**) Unfiltered (SNR = 10 dB). (**h**) Multi-look (SNR = 10 dB). (**i**) Nonlocal (SNR = 10 dB). (**j**) GenInSAR (SNR = 10 dB). (**k**) Φ-Net (SNR = 10 dB). (**l**) Proposed MCAN (SNR = 10 dB).

Since the proposed method achieves the filtering of the multi-channel interferograms at the same time, we change the number of channels to test the performance of the method. Meanwhile, we can compare our proposed method with GenInSAR, and Φ-Net, both of which deal with the single-channel interferogram. Figures 12 and 13 show the results of the filtered interferometric phases with different numbers of channels. Here, we fix the example of the interferogram and randomly select the channels to combine with it. It is clear that both Φ-Net and MCAN (single channel) can suppress the noise well, indicating that the architectures of the network are rationally designed. As for the proposed MCAN with different channels, we find that the denoising performance is better with more channels in general, which shows the effectiveness of the channel attention maps in the proposed architecture.

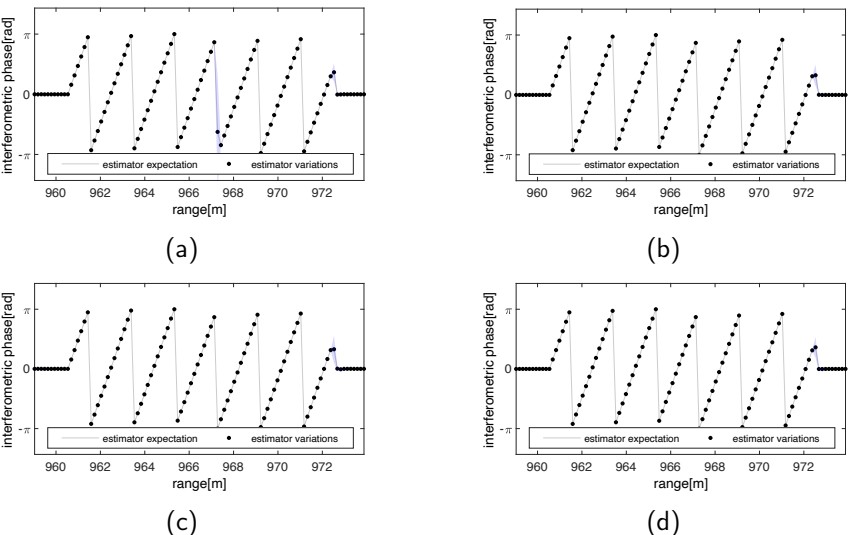

**Figure 12.** Scene.1: The statistical answers for the filtered interferometric phases using our proposed MCAN with different numbers of channels. (**a**) MCAN (single channel). (**b**) MCAN (3-channel). (**c**) MCAN (5-channel). (**d**) MCAN (7-channel).

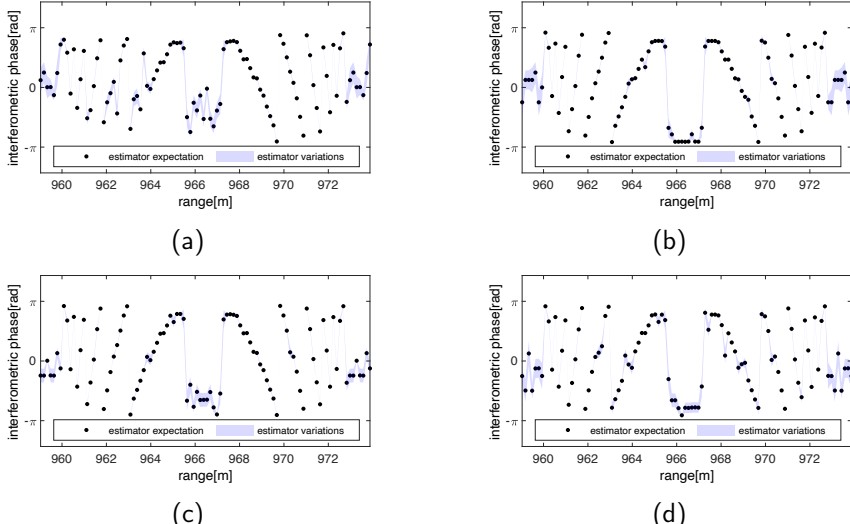

**Figure 13.** Scene.2: The statistical answers for the filtered interferometric phases using our proposed MCAN with different numbers of channels. (**a**) MCAN (single channel). (**b**) MCAN (3-channel). (**c**) MCAN (5-channel). (**d**) MCAN (7-channel).

The complex interferograms are utilized to achieve the tomographic reconstruction. Based on the obtained interferograms, we conduct the inversion of 3-D spatial infor-

mation using TomoSAR technology. The reconstructed 3-D point clouds are shown in Figures 14 and 15. The noise in the multi-channel interferograms results in the outliers in the 3-D reconstruction. Filtering allows us to strongly mitigate this effect. The multi-look filter conducts the operation consisting of averaging pixels in the range and/or azimuth directions, which cannot guarantee the structural properties. Apparently, the network can learn the structure features of the interferograms well, leading to the obvious structures in the reconstruction results. In particular, the reconstructed 3-D point cloud of scene.2 has the excellent structural properties of targets. Meanwhile, the top of the target in scene.2 is preserved well, and the outliers are reduced by the filter. Therefore, our proposed method achieves better performance, leading to great correctness in elevation position.

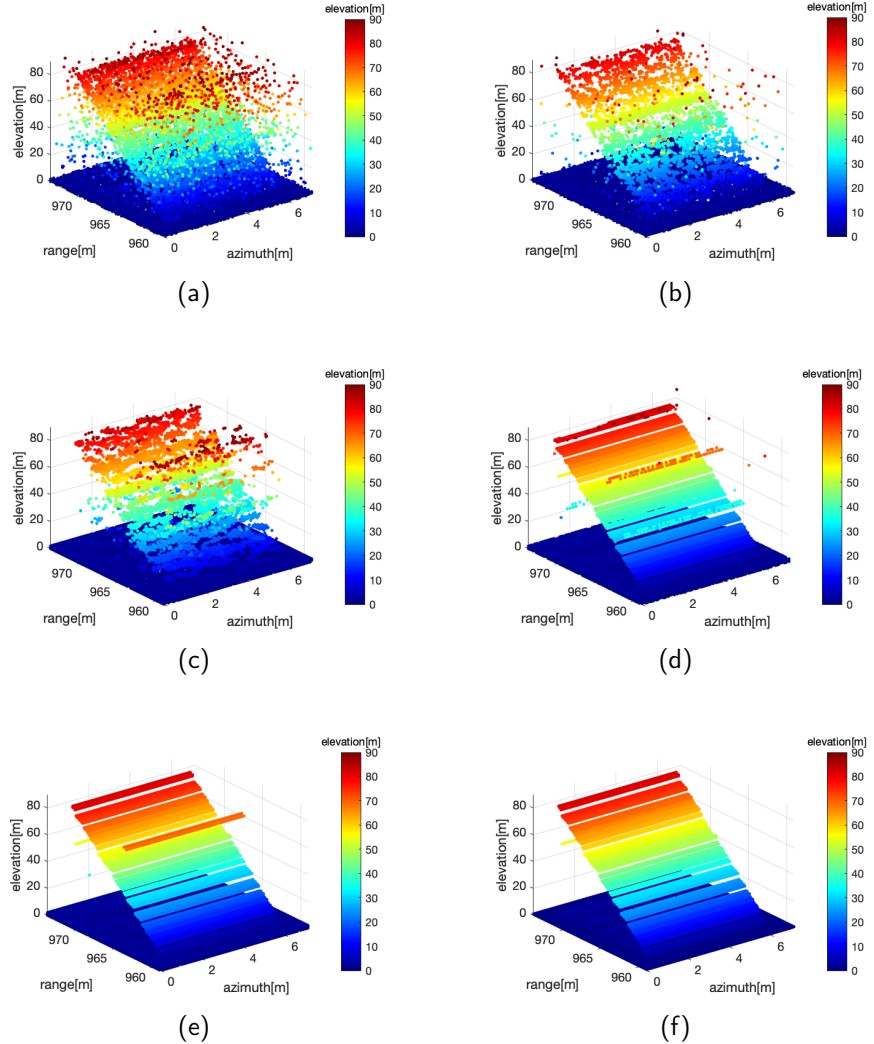

**Figure 14.** Scene.1: The 3-D reconstruction of different filtered results. (**a**) Unfiltered. (**b**) Multi-look. (**c**) Nonlocal. (**d**) GenInSAR. (**e**) Φ-Net. (**f**) Proposed MCAN.

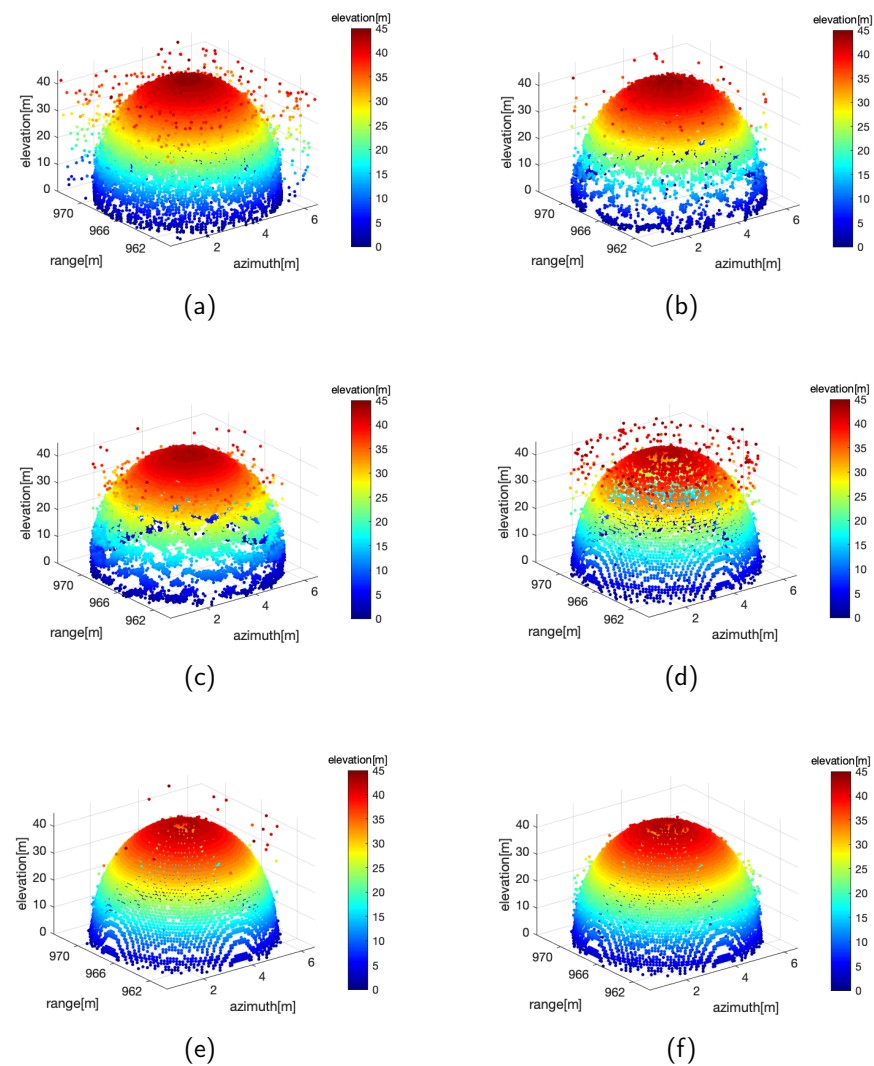

**Figure 15.** Scene.2: The 3-D reconstruction of different filtered results. (**a**) Unfiltered. (**b**) Multi-look. (**c**) Nonlocal. (**d**) GenInSAR. (**e**) Φ-Net. (**f**) Proposed MCAN.

To comprehensively measure the performance of 3-D tomographic reconstruction, we plot the curves of correctness as a function of completeness in Figure 16. The definitions of both correctness and completeness are described in [46,47]. Correctness describes the ratio of the correctly classified points with respect to the total reconstructed points, which increases as the reconstruction has fewer outliers and becomes closer to the ground truth. It is a reflection of the location's accuracy. Completeness, which is often called the detection percentage, denotes the ratio of the correctly classified points with respect to the total points of ground truth. Thus, the larger values of correctness and completeness correspond to the more correct location and fewer holes in the reconstruction results, which move closer to ground truth. It is noted that the outliers would be fewer while the threshold is set larger to form the 3-D point clouds; however, there would be more missing points. Thus, we find that the best trade-off is reached when the noisy multi-channel interferograms are filtered by our proposed method.

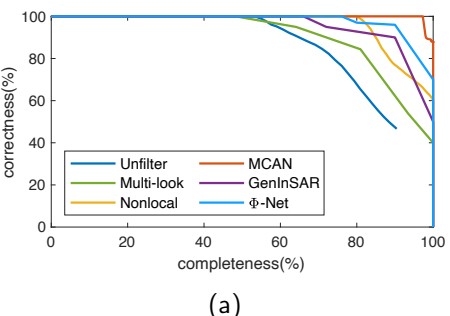 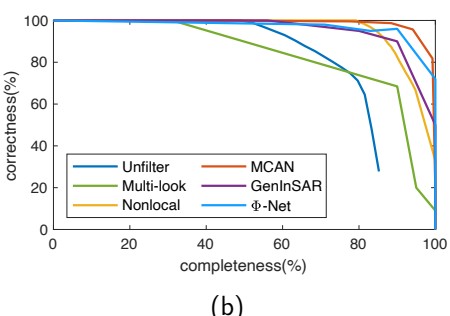

(a)　　　　　　　　　　　(b)

**Figure 16.** The correctness versus completeness to study the performance of 3-D reconstruction based on different filtered results. (**a**) Scene.1. (**b**) Scene.2.

### 3.3. Real Data

In order to verify the performance of the proposed method, the real experimental data acquired in Rizhao city, Shandong province, China, by the Aerospace Information Research Institute, Chinese Academy of Sciences, are utilized to validate our method. The radar system has 16 channels in the cross-track direction. The airborne system flies at an altitude of about 4.3 km above the ground and the local incidence is $39°$. The distance between the adjacent channels is 0.8 m. The corresponding intensity maps of the areas are shown in Figure 17.

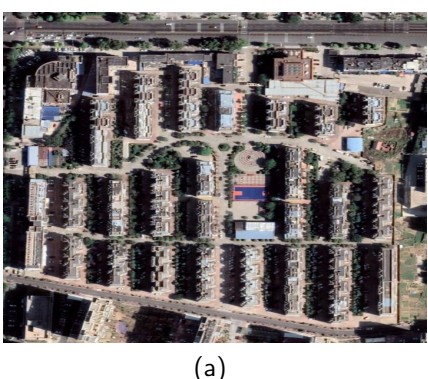 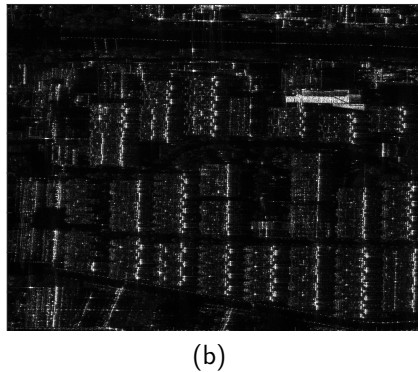

(a)　　　　　　　　　　　(b)

**Figure 17.** (**a**) Optical image of the test site (Google Earth). (**b**) One of the intensity images of the SAR data.

Figure 18 shows the different filtered results using the multi-channel method. It can be seen that there is much phase noise in the unfiltered interferometric phases, which makes it hard to distinguish the details of the targets. Filters can solve this problem by filtering the interferograms. The noise is depressed effectively by the multi-look filter; however, the details are not preserved well. As expected, the proposed method can suppress the outliers well and preserve the structure of the observed scene, especially in the filtered interferometric phase. Here, we calculate the coherence to analyze the denoising performance. As expected, the statistical characteristics of the proposed method present an improvement in coherence, indicating the suppression of decorrelation effects.

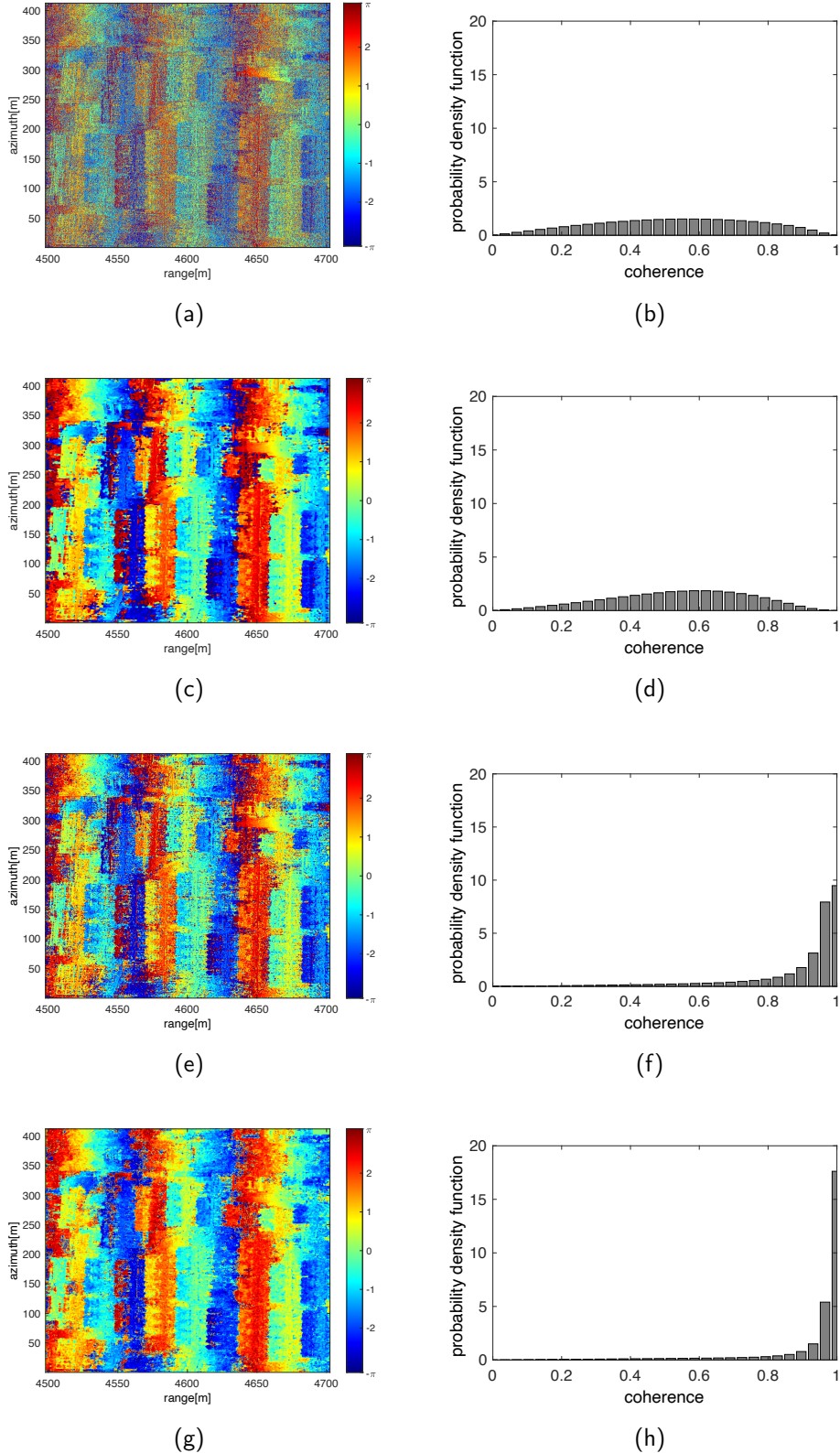

**Figure 18.** The results of interferometric phases and statistical characteristics of coherence based on the different filters. (**a**,**b**) Unfiltered. (**c**,**d**) Multi-look. (**e**,**f**) Nonlocal. (**g**,**h**) Proposed MCAN. From left to right, the images are interferometric phases and statistical characteristics of coherence.

The corresponding tomographic reconstructions are shown in Figure 19. Visual inspection shows that there are plenty of outliers in the tomographic reconstruction of unfiltered SAR data, which can be removed by the proposed method. The building marked by red arrows has a vaguely shaped structure in the reconstruction of unfiltered SAR interferograms, and the detailed shape of the reconstructed target is more clear based on the proposed method, indicating the better completeness of tomographic reconstruction. The outliers would result in poor performance of correctness. Due to the lack of reference data, numerical evaluation of experimental data is difficult. For quantitative analysis of performance, we take the mean of the reconstructed height as the rough estimation of the ground truth. The STD metrics of reconstructed height are calculated, which are 2.25 m, 1.43 m, 1.52 m, and 1.25 m, corresponding to the unfiltered, multi-look filtered, nonlocal filtered, and proposed filtered results. The lower number of outliers brings about the low STD values, indicating the best correctness of the 3-D position. It is clear that the 3-D reconstruction is improved by the proposed filter, which leads to the more correct position of scatterers.

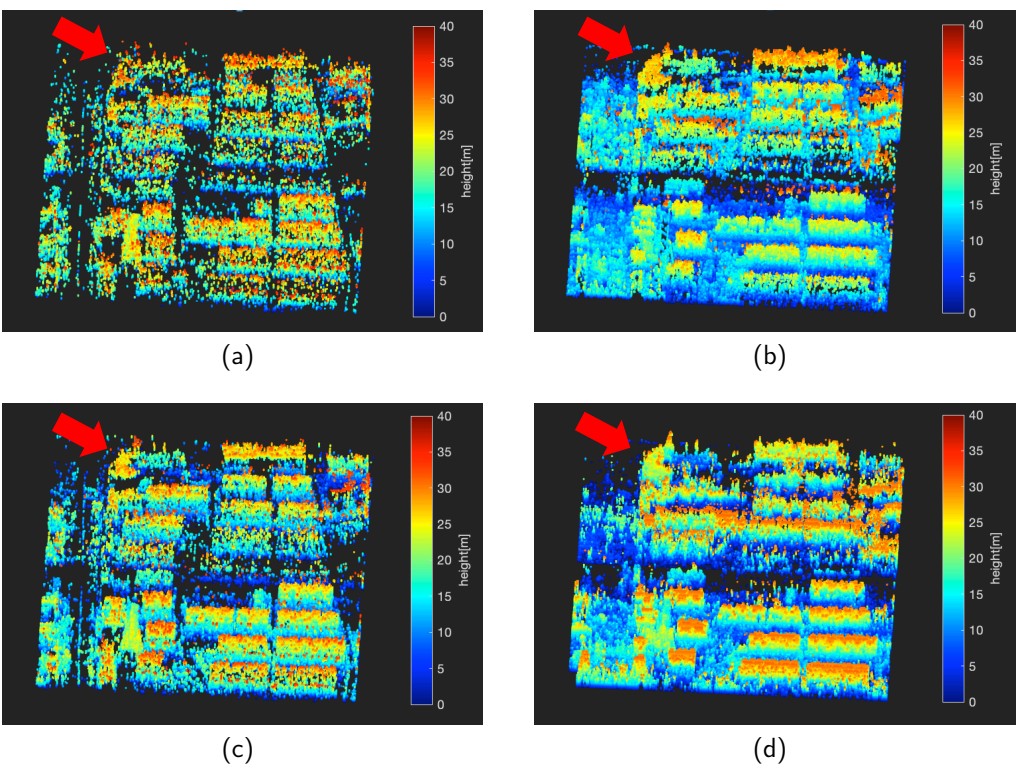

**Figure 19.** The 3-D point clouds of tomographic reconstruction using different filters. (**a**) Unfiltered. (**b**) Multi-look. (**c**) Nonlocal. (**d**) Proposed MCAN.

## 4. Conclusions

In this paper, we proposed a multi-channel attention network to achieve the filtering of multi-channel interferograms, which are applied to TomoSAR. The proposed network utilizes the multi-head mechanism to build the attention blocks, which conduct the local context mixing before exploring the cross-channel information. The former is achieved by the spatial convolution operations in interferograms, to preserve the structure while performing the filtering process. The latter is achieved by the computation of channels attention maps, to make full use of the inter-channel relationship to improve the denoising performance. The network was trained by simulated SAR images and tested by the simulated and real data.

To evaluate the effectiveness of the proposed method, we simulated two scenes to analyze the performances of filtering in different interferometric patterns. The results show that the proposed method can suppress the noisy points in interferograms while preserving the detailed patterns of interferometric phases. The standard deviations of interferometric phases are effectively reduced, which brings about the high correctness of the tomographic reconstruction. Meanwhile, we conducted experiments in different SNRs and numbers of channels to analyze the filtering performance. Moreover, we chose real data to demonstrate the robustness of noise suppression. From the filtered interferograms, we can find that the proposed method has the ability to increase the correctness of the estimated multi-channel interferometric phases. Furthermore, the corresponding reconstructed 3-D point clouds present more correct and complete performances based on the rough ground truth of the tested targets in real data.

It is noted that the proposed method achieves excellent performance in filtering the multi-channel interferograms. The interferograms can be obtained with the same resolution, polarization, and frequency. As such, the filtered interferograms are not time-conservative, which can be utilized in other contexts, such as in the reconstruction of deformation line-of-sight velocity profile along elevation. Thus, our research will continue to find the potential use of this architecture in other contexts in the future.

**Author Contributions:** Methodology, J.L. and B.Z.; Software, J.L. and Z.L.; Validation, J.L.; Formal analysis, J.L. and B.Z.; Resources, B.Z. and Y.W. All authors have read and agreed to the published version of the manuscript.

**Funding:** This research was funded by the National Natural Science Foundation of China Grant No. 61991421.

**Conflicts of Interest:** The authors declare no conflict of interest.

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
