# Peer review of "A Multi-Channel Attention Network for SAR Interferograms Filtering Applied to TomoSAR"

_remotesensing, doi:10.3390/rs15184401_

Round 1
Reviewer 1 Report
This manuscript introduces a multi-channel attention network to filter the multi-channel interferograms for TomoSAR, which consists of multi-channel attention blocks. The simulated and real data experiments demonstrate the effectiveness of the proposed method. There are several comments to be considered:
1. More state-of-the-art methods are suggested to be compared in the simulation. There are only two traditional methods used for the comparison. Other state-of-the-arts, especially deep learning-based ones are not introduced.
2. in Figure 14 (d) The proposed method's result is so clean. Please have a check on the result.
3. The literature review is suggested to be enriched. Maybe split into single-baseline and multi-baseline filtering methods. Some state-of-the-art filtering methods e.g. 10.1109/TGRS.2022.3228279 are better to be summarized.
Author Response
We appreciate the reviewer for the constructive suggestions. The manuscript has been revised according to the comments, and the font of modified parts has been set to be red in the manuscript. The responses to the comments including figures are presented in the attachment.
Please see the attachment.

Reviewer 2 Report
Overview and general recommendation.
This paper elegantly showcases the adept utilization of Synthetic Aperture Radar (SAR) methodology alongside its pivotal applications in diverse fields. The technique's profound significance is underscored by its inherent precision, offering a multitude of indispensable applications. Drawing upon over two decades of personal immersion in RADAR/SAR data, I hold a profound appreciation for the paper's content. However, as a committed scientist, it remains imperative to objectively assess and candidly address the nuances of the material at hand.
The abstract requires significant improvement as it currently struggles to effectively encapsulate the paper's ideas and concepts. The language usage, particularly in English, poses some challenges. Engaging a native English speaker for thorough review is highly advisable and should be prioritized. The current abstract appears unrefined, resembling an initial draft. It would greatly benefit from a more deliberate consideration of the paper's contents, highlighting the core accomplishments attained. A revised abstract should be aimed at succinctly conveying the primary outcomes of the study, distinct from an introduction, while demonstrating a higher level of articulation. I encourage a comprehensive reiteration of this section.
The Introduction appears to be lacking in fairness and could benefit from significant improvements. In specific sections, notable corrections are required to ensure accuracy and clarity. While a complete rewrite of the Introduction is a suggestion worth considering, it is not an absolute requirement. It is evident that there is a notable deficiency in the inclusion of essential references, and I urge you to address this oversight promptly. Your attention to rectifying these aspects will contribute to the overall quality and rigor of the Introduction.
Main Question Addressed by the Research:
The main question addressed by the research is not explicitly mentioned in the paper. It would be helpful to have a clear statement of the primary research question that the paper seeks to answer. This could be added to the Introduction section, ideally in the form of a research objective or hypothesis.
Originality and Relevance of the Topic:
You acknowledge the paper's importance and relevance due to the adept utilization of Synthetic Aperture Radar (SAR) methodology and its applications across diverse fields. It would be beneficial to expand upon why you find the topic original and relevant, perhaps by mentioning how SAR methodology contributes to advancements in existing research or solves specific challenges in the field.
Contribution Compared to Other Published Material:
Elaborate on how this experience enriches the paper and differentiates it from other published materials. Specify which accomplishments and findings are particularly novel. Methodology Improvements: While you don't go into detail about specific issues in the methodology, your feedback suggests there are areas for improvement. In the methodology section, provide specific suggestions for enhancements or clarifications. You might consider recommending additional validation techniques, discussing potential sources of error, or proposing ways to strengthen the method's robustness.
Consistency of Conclusions with Evidence:
You should ensure that the conclusions drawn in the paper align with the evidence and arguments presented. If there are discrepancies or gaps, they should be addressed either by revising the conclusions or providing additional analysis. Appropriateness of References: You've highlighted a deficiency in the inclusion of essential references. In the Introduction and throughout the paper, provide specific examples of missing references that are pertinent to the discussion. Suggest relevant works by other researchers that could bolster the paper's foundation and demonstrate a comprehensive understanding of the field.
Tables and Figures:
I've expressed dissatisfaction with the quality of the figures. It's crucial to upgrade the figures to enhance the paper's visual appeal and clarity. If possible, provide specific recommendations for improving each figure, such as using higher resolution images, improving labeling, or presenting data in a more visually engaging way. You can also address the issues you've identified in the end of the Introduction.
Pls improve the quality of Figs> they are horrible; unacceptable (they look like a screenshot!!). The end of the Introduction is very bad. Change them.
Results, and discuses, need to be reconsidered a bit; BUT, overall they are OK.
I like this paper very much: good experiments have been done; however, I think this work must be improved, and lots of things to do; agreed? I think the paper is very raw, and must be improved tremendously.
Engaging a native English speaker for thorough review is highly advisable and should be prioritized
Author Response
We appreciate the reviewer for the valuable suggestions and comments. The manuscript has been revised, and the font of modified parts has been set to be red in the manuscript. The responses to the comments are presented in the attachment.
Please see the attachment.

Reviewer 3 Report
The paper is really interesting and potentially it could have a great impact. However, somehow, the description of the applied methods is confusing and lots of things are assumed known to the readers. I would suggest improving the description of the used network providing additional details on its implementation and more references to past works exploiting the same method. The paper relies on the noise filtering of SAR interferograms and considers the TomoSAR framework, directly. As far as I understand, the same method can be applied to sequences of conventional SAR interferograms. The need of using TomoSAR needs to be better contextualized (even though it's clear for expert readers). The performance of the method with respect to conventional multi-looking and also non-local methods is proposed. I am wondering whether the obtained interferograms are time-conservative or not. The method is applied to the whole stack of interferograms but is the time consistency among (for instance) triplets of interferograms considered? Conventional multi-look interferograms filtered independently do not consider time consistency. It's a curiosity of mine to know whether this issue is automatically taken into consideration by the used network (or not). This could have some potential implications considering its application in a context non-necessarily related to tomography. Finally, the conclusion section is too limited. The authors must provide explanations on the potential use of this method in other contexts and explain better the role of the achieved results in a more general framework.
I'm not qualified to judge the English level and style. Anyway, the paper can be read easily and I did not find any significant mistakes.
Author Response
We appreciate the reviewer for the constructive suggestions and comments. The manuscript has been revised, and the font of modified parts has been set to be red in the manuscript. The responses to the comments are presented in the attachment.
Please see the attachment.

Round 2
Reviewer 2 Report
After conducting a comprehensive review of both the original manuscript submitted to Remote Sensing and its subsequent revised version, I am delighted to highlight the remarkable enhancements that have been made. The modifications incorporated in the new iteration have undeniably captivated my attention and garnered my resounding approval. Thus, I am pleased to announce my acceptance of the revised manuscript and express my utmost satisfaction with the revisions implemented.
After conducting a comprehensive review of both the original manuscript submitted to Remote Sensing and its subsequent revised version, I am delighted to highlight the remarkable enhancements that have been made. The modifications incorporated in the new iteration have undeniably captivated my attention and garnered my resounding approval. Thus, I am pleased to announce my acceptance of the revised manuscript and express my utmost satisfaction with the revisions implemented.